# SING: A Plug-and-Play DNN Training Technique

## Abstract

We propose SING (StabIlized and Normalized Gradient), a plug-and-play technique that improves the stability and generalization of the Adam(W) optimizer. SING is straightforward to implement and has minimal computational overhead, requiring only a layer-wise standardization of the gradients fed to Adam(W) without introducing additional hyper-parameters. We support the effectiveness and practicality of the proposed approach by showing improved results on a wide range of architectures, problems (such as image classification, depth estimation, and natural language processing), and in combination with other optimizers. We provide a theoretical analysis of the convergence of the method, and we show that by virtue of the standardization, SING can escape local minima narrower than a threshold that is inversely proportional to the network's depth.

## 1 Introduction

Neural network training is a highly non-convex and stochastic optimization problem, complicated by hidden dynamics between the optimization algorithm and the network architecture. Several common pitfalls have been identified, such as bad initialization, vanishing and exploding gradients [3, 32], abrupt shifts in the distribution of layer inputs (the so-called internal covariate shift [19]). Significant progress has been made by tackling these issues either by architectural improvements [1, 17] or with better optimizers [21, 28, 39].

The Adam(W) optimizer [21, 26] is widely adopted for neural network training due to its ability to combine first and second-order moments of the gradient, mitigating the sensitivity to the learning rate, and providing adaptability to gradient updates of different magnitude or sparsity. It is applicable to widely different architectures, from convolutional to transformers, and application domains. Nonetheless, it has shown instabilities in specific scenarios, such as large-scale problems [5, 29] or, as we demonstrate in this work, some image-to-image tasks. These instabilities manifest as spikes in the training loss which might involve a prolonged recovery periods - if it recovers.

**Contributions.** In this work we propose a simple layer-wise gradient standardization as a technique to improve the stability of existing optimizers. Our technique, SING, is plug-and-play: by simply changing the gradient fed to AdamW (or any other "host" optimizer) it integrates seamlessly without introducing any additional hyperparameters. As such, it does not require any additional fine-tuning apart from that of the host optimization framework. In this way, SING preserves the desirable properties of the host optimizer but with increased stability.

We notably theoretically show that the optimizer is capable to escape narrow local minima *within a single step*, given a sufficiently high learning rate (see Theorem 3.1). Moreover, the magnitude of this learning rate is *inversely proportional* to the depth of the network *i.e.* for a fixed learning rate, the higher the number of layers of the network, the lower the learning rate must be to escape local minima. This highlights the compatibility of our technique with deep neural networks. Since narrow local minima are often associated with poor generalization in non-convex optimization landscapes

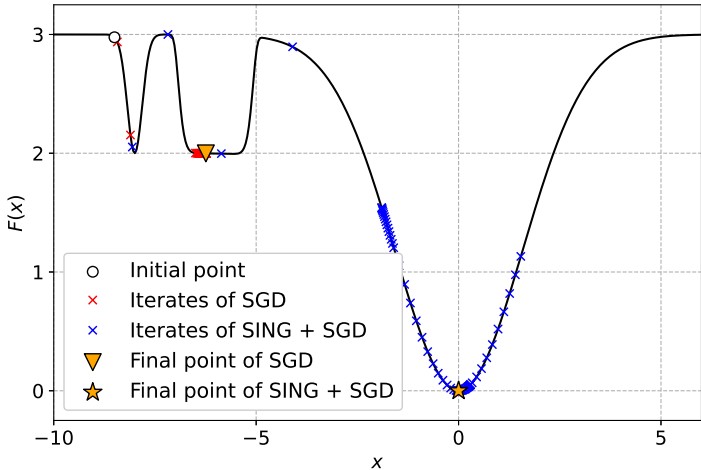

Figure 1: Optimization of the function with three local minima. A gradient descent (noted SGD) with the proposed gradient standardization SING can escape narrow local minima (Theorem 3.1). SING steps jump over narrow local minima in one step. Conversely, SGD without SING steps jump over the first local minimum but stays blocked in the second one because the gradient amplitude is too small. The learning rate is reduced using a cosine decay.

[6, 15, 16, 20, 33], it is crucial for an optimizer to avoid such pitfalls. We capitalize on this theoretical result and stabilize the gradient using several techniques [18, 57, 40] to reach a step size as large as possible, and thus escape from as many local minima as possible.

Although our initial focus was to address Adam's limitations, our technique demonstrates robust generalization even in tasks where Adam performs well. Notably, we show improvements over AdamW on several tasks such as image classification, depth estimation and natural language processing.

The paper is organized as follows. Section 2 describes the proposed method, followed by its theoretical analysis in Section 3 where the properties of SING are presented. We discuss related work in Section 4. In Section 5 we provide extensive numerical experiments which demonstrate the effectiveness of the proposed method on a variety of tasks in natural language processing and computer vision.

## 2 Algorithm

We seek to approximate the solution to the following optimization problem,

$$\min_{x \in \mathbb{R}^p} F(x). \tag{1}$$

We assume there exists a random function $f : \mathbb{R}^p \to \mathbb{R}$ such that $\mathbb{E}[\nabla f(x)] = \nabla F(x)$ for all $x \in \mathbb{R}^p$, and that we have access to an oracle providing i.i.d. samples $(f_n)_{n \in \mathbb{N}}$ [9].

In the case of neural networks, the optimization variable $x$ represents the parameters of the network, $F$ is the oracle loss and $f$ is the empirical loss evaluated on a random mini-batch. The parameters of a neural network have a specific structure. They are made of the concatenation of the parameter tensors from each layer of the network. We use $D \in \mathbb{N}$ to denote the number of these parameters tensors and define $(I_k)_{k \in [\![1,D]\!]}$ such that $x_{I_k} = \{x_i : i \in I_k\}$ represents the $k$-th parameter tensor. As an example, let's consider the neural network $\mathcal{N}(\cdot; A, b) = A \cdot + b$. In this case, the network has two parameter tensors, hence $D = 2$. The first parameter tensor is $x_{I_1} = A$ and the second is $x_{I_2} = b$.

Our objective is to endow an optimizer with several key properties: **1)** stability **2)** capacity to escape narrow local minima **3)** adaptability to geometry of the energy landscape, and **4)** convergence. Importantly, we want to achieve all of these properties without adding any additional hyper-parameters and with minimal computational overhead. We achieve the first property – stability – by dividing the gradient by its norm. This prevents vanishing and exploding gradients, which can lead to unstable training. This also allows to use fixed gradient steps, which enables the algorithm to move past narrow local minima, as we show in the next section.

```python
def optim_step(model, lr, beta, weight_decay, ε):      def centralize(grad):
  for p in model.parameters():                           if grad.dim() > 1:
    # Standardization                                        dims = tuple(range(1, grad.dim()))
    p.grad = centralize(p.grad)                             mean = grad.mean(dims, keepdim=True)
    p.grad = normalize(p.grad, ε)                          grad = grad - mean
    # Weight decay                                         return grad
    p = p * (1 - lr * weight_decay)
    # Optimizer                                          def normalize(grad, ε):
    update = optimizer(p.grad, beta)                       grad = grad / (grad.norm() + ε)
    # Parameter update                                     return grad
    p = p - lr * update
```

Algorithm 2: PYTORCH implementation of our algorithm. The $\Gamma$ operator is implemented by NORMALIZE and $\phi$ by CENTRALIZE. Our technique can be used within any existing first order method *i.e.* the OPTIMIZER function can be any optimizer (see Table 3 for a comparison).

We define the steps taken by our optimizer by

$$x_{t+1} = x_t - \eta \frac{\phi(\nabla f(x_t))}{\Gamma(\phi(\nabla f(x_t)))}, \tag{2}$$

where $\phi$ is the gradient centralization operation [49] and the division is applied element-wise. The operator $\Gamma$ corresponds to the parameter-wise normalization *i.e.*

$$\Gamma(x)_i = \|x_{I_k}\|_2, \quad \text{where } k \in [\![1, D]\!] \text{ and } i \in I_k. \tag{3}$$

In theory, there could be a division by zero in (2). To avoid this we can add $\epsilon = 10^{-8}$ to the denominator although it is not strictly necessary because the gradient norm is large in practice. This parametrization naturally arises in usual Deep Learning frameworks, see Algorithm 2 for the PYTORCH implementation.

Our setting differs from regular normalized gradient descent in two ways: we center the gradients before normalizing and we perform the normalization on a parameter-wise basis. This is particularly important for large networks where the norm of the full gradient can be very large, making it nearly impossible to train the network effectively.

## 3 Theoretical Analysis

This section analyzes the key properties of our technique. Theorem 3.1 demonstrates how normalization techniques aid in escaping local minima. Theorem 3.2 establishes stability results, including several invariance properties of the algorithm. Moreover, Theorems 3.3 and 3.4 provide insights into the rate of convergence of our algorithm in a stochastic setting, under mild assumptions. For complete proofs and technical details, please refer to Appendix A.

### 3.1 Escaping from narrow local minima

One of the key properties of our algorithm is its ability to escape from narrow local minima. This is crucial because the stochasticity of the optimization landscape often leads to the creation of artificial local minima, generally associated with poor generalization performance [6, 15, 16, 20, 33]. To achieve this we normalize the gradient to take fixed-size steps during training, where the learning rate controls the step size. Doing so allows the escape from narrow local minima provided the steps are large enough. This property is central to our algorithm and leads to better generalization performance.

For simplicity, we assume a deterministic setting in this section. We show that the normalization procedure helps the optimizer to escape narrow local minima. To formalize this observation, we first define the *basin of attraction* of a critical point of $F$.

**Definition 3.1.** Let $x^*$ be a critical point of $F$. The basin of atraction of $x^*$ is defined to be the set $W(x^*)$ such that

$$W(x^*) \stackrel{\text{def}}{=} \{x \in \mathbb{R}^p : \langle \nabla F(x), x - x^* \rangle \geq 0\}.$$

Moreover, we write $\mathcal{B}(x^*)$ to be the largest ball contained within $W(x^*)$, and $r$ its radius.

96    In the previous definition, if $x^*$ is a saddle point, $A(x^*) = \{x^*\}$ and $r = 0$.

97    **Theorem 3.1** (Escaping from narrow local minima). Let $x_t$ be the sequence of iterates defined by (2)
98    and $y_t$ the sequence of iterates of gradient descent,

$$y_{t+1} = y_t - \eta_{\text{GD}} \nabla F(y_t). \tag{4}$$

99    Assume that $x_t \in \mathcal{B}(x^*)$ (resp. $y_t \in \mathcal{B}(x^*)$) *i.e.* the ball contained in the basin of attraction of $x^*$,
100   defined in Definition 3.1. Also, assume that $x_t$ (resp. $y_t$) is not a critical point *i.e.* $\nabla F(x_t) \neq 0$ (resp.
101   $\nabla F(y_t) \neq 0$). If the stepsize is sufficiently large,

$$\eta_{\text{SING}} \geq \frac{2r}{\sqrt{D}}, \qquad \eta_{\text{GD}} \geq \frac{2r}{\|\nabla F(y_t)\|_2}, \tag{5}$$

102   then the iterates $x_{t+1}$ (resp. $y_{t+1}$) is outside the set $\mathcal{B}(x^*)$. See Figure 1 for an illustration.

103   We see that GD struggles to escape local minima: under mild assumptions on $\nabla F$, the closer $y_t$ is
104   to $x^*$ the higher the learning rate must be to escape from $A(x^*)$. Indeed, for GD there is no finite
105   step-size $\eta_{\text{GD}}$ that guarantees escaping $A(x^*)$. In contrast, Theorem 3.1 tells us that our algorithm
106   escapes $A(x^*)$ in a single step, provided the learning rate is sufficiently large. Furthermore, as the
107   number of parameter tensors in the model increases, it becomes easier to escape from $A(x^*)$. This is
108   an important advantage of our algorithm over GD, especially for large models where the optimization
109   landscape can be highly complex and difficult to navigate.

110   When the Hessian at $x^*$ is well conditioned, escaping from $A(x^*)$ is roughly equivalent to escaping
111   from the local minimum. Therefore, it is crucial to use the highest possible learning rate. However,
112   using a high learning rate can be problematic as the gradients are unstable and tend to oscillate leading
113   to suboptimal convergence. To address this issue, several methods have been proposed to stabilize the
114   gradients and allow for larger learning rates. Such methods include gradient centralization, LookA-
115   head [54], different momentum strategies such as Adam [21], AdaBelief [57] or even AdaFactor
116   [37] and larger batch sizes, among others. For this reason, the final implementation of our algorithm
117   incorporated within AdamW features LookAhead and softplus calibration [40]. Note however that it
118   does not introduce any additional hyper-parameters as the parameters of these stabilization methods
119   are fixed once and for all.

## 3.2   Invariance properties

121   In this section, the setting is considered deterministic for simplicity. This section examines the
122   invariance properties of the technique.

123   Firstly, we show that a rescaling of the objective function,

$$\min_{x \in \mathbb{R}^p} \tilde{F}(x) \stackrel{\text{def}}{=} \alpha F(x), \quad \alpha > 0. \tag{6}$$

124   does not affect the updates. This property is desirable as the network's performance is unaffected
125   by a scaling of the loss. A similar invariance property applies to changes during training that cause
126   a rescaling of the gradients of a layer. If during training, the output of one layer of the network is
127   rescaled, it won't affect the update of the previous layers, thus allievating part of the problem of
128   internal covariate shift [19].

129   Second, the algorithm presented in this paper preserves the mean *i.e.*

$$\sum_{i=1}^{p} [x_{t+1}]_i = \sum_{i=1}^{p} [x_t]_i, \tag{7}$$

130   where $[x]_i$ corresponds to the $i$-th component of the vector $x$.

131   **Theorem 3.2.** The iterates defined by (2) are invariant w.r.t. transformation (6), and preserve the
132   mean (7).

133   The property of preserving the mean has been demonstrated to improves the stability of the optimiza-
134   tion process in deep neural networks [49]. Moreover, it is motivated by the observation that many
135   non-linear layers demonstrate a mean-shift behavior [19], which alters their behavior based on the
136   sign of input values. This mean-shift behavior is mitigated by the presence of normalization layers,
137   that re-scale and shift the weights. Preserving the mean enhances the stability of the optimization
138   dynamics when normalization layers are present.

Furthermore, normalizing the centered gradients mitigates a potential pathological scenario where the gradient signal is diminished. Indeed, the mean of the gradient can hinder the important signal when the mean is too large compared to the centered gradient [49]. However, in such case the amplitude of the centered gradient can be relatively small, preventing efficient updates. Normalizing the gradient solves this issue by preserving its amplitude.

## 3.3 Convergence

In this section, two theorems of convergence are provided. In the first one, the normalization is studied without the centralization. Under mild assumptions, we show the $\ell^2$-norm of the gradient can be reduced to any desired precision. In the second one, we consider the full setting and show the same result for the $\phi$-norm (which is a pseudo-norm). We assume that the stochastic gradient has a $\sigma$-bounded variance ($\sigma > 0$) *i.e.*

$$\forall x \in \mathbb{R}^p, \mathbb{E}\left[\|\nabla F(x) - \nabla f(x)\|_2^2\right] \leq \sigma^2, \tag{8}$$

and the objective function $F$ is positive and $L$-smooth,

$$\forall x, y \in \mathbb{R}^d, \|\nabla F(x) - \nabla F(y)\|_2 \leq L\|x - y\|_2. \tag{9}$$

**Theorem 3.3** (Convergence without gradient centralization). Let assumptions (8) and (9) hold. Assume the gradient is computed across a mini-batch of size $B = \frac{\sigma^2}{\epsilon^2}$. Let $x_t$ be the sequence of iterates (2) with $\phi = I$. Then, we have

$$\frac{1}{T}\sum_{t=0}^{T-1}\mathbb{E}[\|\nabla F(x_t)\|_2] \leq \frac{F(x_0)}{\eta T} + (1 + \sqrt{D})\epsilon + \frac{\eta LD}{2}. \tag{10}$$

If we set $\tau \sim \mathcal{U}([\![0, T-1]\!])$, $\eta = \frac{2\epsilon}{L}$ and $T = \frac{LF(x_0)}{2\epsilon^2}$, we obtain $\mathbb{E}[\|\nabla F(x_\tau)\|_2] \leq (2 + \sqrt{D} + D)\epsilon$. Therefore, the iteration complexity and computation complexity to achieve an $\epsilon$-stationary point are $\mathcal{O}(1/\epsilon^2)$ and $\mathcal{O}(1/\epsilon^4)$, respectively.

**Theorem 3.4** (Convergence with gradient centralization). Let assumptions (8) and (9) hold. Assume the gradient is computed across a mini-batch of size $B = \frac{\sigma^2}{\epsilon^2}$. Let $x_t$ be the sequence of iterates (2). Then we have

$$\frac{1}{T}\sum_{t=0}^{T-1}\mathbb{E}[\|\nabla F(x_t)\|_\phi] \leq \frac{F(x_0)}{\eta T} + (1 + \sqrt{D})\epsilon + \frac{\eta LD}{2}, \tag{11}$$

where $\|\cdot\|_\phi^2 = \langle \cdot, \phi(\cdot)\rangle_2$ is a pseudo-norm. If we set $\tau \sim \mathcal{U}([\![0, T-1]\!])$, $\eta = \frac{2\epsilon}{L}$ and $T = \frac{LF(x_0)}{2\epsilon^2}$, we obtain $\mathbb{E}[\|\nabla F(x_\tau)\|_\phi] \leq (2 + \sqrt{D} + D)\epsilon$. Therefore, the iteration complexity and computation complexity to achieve an $(\epsilon, \phi)$-stationary point are $\mathcal{O}(1/\epsilon^2)$ and $\mathcal{O}(1/\epsilon^4)$, respectively.

Note that Theorem 3.4 only gives us an $(\epsilon, \phi)$-stationary point *i.e.* in the limit $\epsilon \to 0$, $\nabla F(x_\tau)$ converges to a point in $\text{Ker}(\phi) = \text{Span}((1, \ldots, 1)^T)$. Indeed, applying $\phi$ to the gradients amounts to do a projected gradient descent onto the set of weights with the same mean as the initial weights.

We argue that this "weaker" convergence result it is not problematic. Reaching a point in $\text{Ker}(\phi)$ means that the optimization process cannot go any further without violating the constraint. However, since neural networks have lots of parameters, adding one constraint to the solution is not likely to lead to worse performance [49].

## 4 Related Work

The most-used optimizer nowadays is Adam [21], which computes the *entrywise* first and second order moments of the gradient, and uses them to adaptively normalize the gradient. In contrast, SING first removes to the gradient its mean and divides it by its norm (standardization) prior to any further computations *at the layer level*. Furthermore, in Adam the first and second orders are averaged temporally while ours is not. Numerous variants of Adam have been proposed, mainly focusing on stabilizing the iterations of Adam. Notably, the popular AdamW [26] optimizer corrects how weight decay is applied in Adam, yielding a more robust method training larger models, for instance for

|  |  | SGD | AdamW [26] | W+GC [49] | AdaBelief [57] | W+SING |
|---|---|---|---|---|---|---|
| ImageNet | ResNet18 | 72.44% | 72.58% | 72.31% | 72.49% | **72.93%** |
| ImageNet | ResNet34 | 75.36% | 75.29% | 75.26% | 75.49% | **75.67%** |
| CIFAR100 | ResNet18 | 75.63% | 77.95% | - | - | **78.24%** |

Table 1: Top-1 accuracy classification results of ResNet [17] on CIFAR100 [22] and ImageNet [10] when trained from scratch. W+GC stands for the combination of AdamW [26] and Gradient Centralization [49] and W+SING stands for AdamW and SING. For CIFAR100, the results are averaged across five runs. The standard deviations (reported in the appendix) are sufficiently low to say that W+SING has a clear edge on AdamW. We do not report the mean and standard deviation on ImageNet-1K because we only launched each experiment once due to computational constraints but we can reasonably expect our results to be significant (see Figure 3 for a motivation).

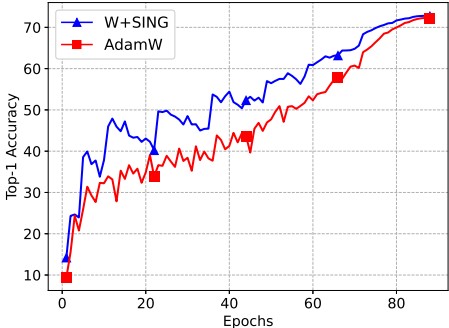
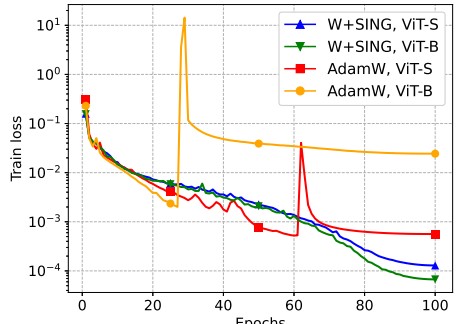

Figure 3: Evolution of the accuracy throughout training on ImageNet-1K with a ResNet18. The almost-periodic oscillation of the metric is typical of SING, and could be explained by the large steps taken by the optimizer. As illustrated in Figure 1, at the beginning the learning rate is very high to avoid local minima and is slowly reduced to reach convergence.

Figure 4: An illustration of a failure case of Adam when trained on ViT-small. The train loss suddenly spikes during training, reducing the final performance. The learning rate scheduler is a cosine decay, hence the learning rate is already small when the explosion occurs.

training (visual) Transformers in practice [41]. Also in the panel of corrections to Adam, RAdam [24] proposes to fix the variance of adaptive learning rate by rewriting the update term, AdaBound [27] and AdaMod [12] clip the update, and AdaNorm [14] directly clips the gradient instead. Conversely, EAdam [52] and SAdam [40] target improving the $\epsilon$ term in the denominator. AdaBelief [57] is another variant of Adam. It computes an estimate of the standard deviation instead of the second order moment. AdaFactor [37] factorizes the elements of the gradient to reduce the memory consumption of the optimizer. The authors propose as well an analysis of the instabilities of Adam, and fixes. In this work, we also target reducing Adam's instabilities via gradient standardization.

The works most closely related to ours are LARS [50] and LAMB [51]. Indeed, both optimizers normalize the gradient in a layer-wise fashion like us. However, both methods multiply the normalized gradient by the weight norm. This multiplication is undesired in our case as it would tame down our main theoretical result in Section 3 (Theorem 3.1) which is central to our work. Indeed, this theorem is the keystone to building a stable optimizer able to escape from narrow local minima using larger learning rates, whereas these methods leverage very large batch size to improve performance. Additionally, our method is hyperparameter-free in contrast to those of [50, 51]. Furthermore, these methods are new optimizers to be used as a replacement for Adam(W) whereas SING is a technique that can be used within any optimizer.

Other approaches leverage standardization to better train neural networks: Weight Standardization [34] and Weight Normalization [18, 36] parameterize the weights of the network to allow for a smoother training. While this affects the gradients, this approach is orthogonal to ours and could be used with our technique.

| | Maximum LR | AdamW [26] | AdamW + SING |
|---|---|---|---|
| ViT-S [13] | 0.05 | 78.13% | 96.56% |
| ViT-S [13] ($\pm$ Normalization) | NC | 93.00% (+14.87%) | NC |
| ViT-S [13] ($\pm$ GC [49]) | 0.01 (1/5) | 77.46% (-0.67%) | 93.86% (-2.70%) |
| ViT-S [13] ($\pm$ LookAhead [54]) | 0.01 (1/5) | 54.79% (-23.35%) | 95.63% (-0.93%) |
| ViT-S [13] ($\pm$ Softplus [40]) | 0.005 (1/10) | NC | 89.38% (-7.18%) |
| ViT-B [13] | 0.05 | NC | 97.15% |

Table 2: Ablation study of the different components of SING using a ViT-S on RDE. The reported metric is the accuracy. Each component allow for a higher learning rate. For AdamW, we added the component and studied the convergence. For AdamW + SING, we removed it. As the theory suggests, the higher the learning rate, the higher the final performance. NC stands for no convergence *i.e.* the loss could not be stabilized throughout the iterations. The maximum LR reported corresponds to the one for AdamW + SING. As displayed in Figure 4, the training of ViT-B spiked resulting in irrelevant performance. Note that for AdamW, the very unstable nature of the training largely widens the gaps in performance when adding a component. In all cases, the best learning rate using AdamW alone was $10^{-3}$.

Another part of the literature focuses on improving the stability of training processes to ensure smoother convergence. Notably, techniques such as LookAhead [54] adopt an approach where weights computed over the previous $k$ iterations are averaged. Similarly, Gradient Centralization [49] involves subtracting the mean of the gradient, effectively reducing its $\ell^2$ norm. In our work, we draw upon these techniques, but it is important to highlight that our approach is distinct and independent from this line of research.

Lastly, it is common in non-convex optimization to normalize the gradient descent algorithm [8, 30, 56]. This line of work supports that the standardization strategies is a simple way to find a better minimizer. In this work, we translate this strategy to deep learning.

# 5 Experiments

In this section, we evaluate SING on classification, depth estimation and natural language processing. We run all the experiments on a single Tesla V100 GPU with 32GB of VRAM. The code to reproduce the results will be made available upon publication.

## 5.1 Image classification

We evaluate our technique on the large-scale ImageNet-1K dataset [10] which consists of 1.28 million images for training and 50K images for validation from 1000 categories. We use the FFCV library [23] and its recipe: the data augmentation consists in random horizontal flips and random resized crops. Notably, the downsampling layers are replaced by BlurPool [55]. The size of the images is $192 \times 192$ during training and $224 \times 224$ at evaluation [42]. Our networks are trained for 88 epochs with a batch size of 1024. The loss to be minimized is the cross-entropy with a label smoothing [38] of 0.1. For all networks, there is a 5-epoch linear warmup and a cosine decaying schedule afterward.

We carefully design our hyper-parameter tuning strategy to ensure a fair comparison. First, we tune the learning rate among limited values: $\{5 \times 10^{-4}, 10^{-3}, 5 \times 10^{-3}, 10^{-2}\}$ for AdamW and $\{5 \times 10^{-3}, 10^{-2}, 5 \times 10^{-2}, 10^{-1}\}$ for SING used within AdamW. In the rare cases where the best learning rate found is one of the extreme value of the set, additional learning rates were tested. For all networks and optimizers the best learning rate found is the last one before the training explodes. Then, we tune the weight decay using the best learning rate found. The values assessed for the weight decay are $\{5 \times 10^{-4}, 5 \times 10^{-3}, 5 \times 10^{-2}, 5 \times 10^{-1}\}$. Finally, the results are reported in Table 1. We notice that SING combined with AdamW systematically outperforms AdamW. The evolution of the accuracy throughout training can be seen in Figure 3. SING seems to outperform AdamW during the entire training, but seem to loose its edge at the end of the training. We leave the study of this phenomena for future works.

|          | SGD             | AdamW [26]        | AdaBelief [57]    | AdaFactor [37]   |
|----------|-----------------|-------------------|-------------------|------------------|
| w/o SING | 0.25%           | 78.13%            | 60.26%            | 74.98%           |
| w/ SING  | 94.25% (+94.23%) | 96.56% (+18.43%) | 96.70% (+36.44%) | 76.26% (+1.28%) |

Table 3: Combination of SING with other optimizers for training a ViT-S [13] model on RDE. Note that SGD barely works on this task and model despite the hyper-parameter tuning. We argue that its performance could be further improved by tuning the momentum hyper-parameter. See the Appendix for more details. We notice that for three out of four optimizers, incorporating SING helps improve the performance. See the Appendix for more details.

Additionally, we trained a ResNet18 on CIFAR100 [22], which consists of 50K training images and 10K testing images from 100 classes. The images are of size $32 \times 32$. The network was trained for 300 epochs using a batch size of 128. The learning rate scheduler and the tuning strategy are the same than for ImageNet. The results are visible in Table 1. We see that even in this challenging setting, the combination of AdamW and SING outperforms AdamW and SGD.

## 5.2 Depth Estimation

In this section, we investigate the performance of our optimizer on a depth estimation task using a synthetic dataset. The RDE dataset [7] consists of 50K $128 \times 128$ images of rectangles randomly placed within an image. Depth naturally arises as rectangles are placed on top of each other. The goal is to predict the depth for each pixel in the image, depending on which rectangle it belongs too. This task is interesting because although there exists a simple algorithm that computes the desired depth with 100% accuracy, neural networks struggle to get good performance. Notably, we found training a ViT-small [13] on this task in an image-to-image fashion to be particularly challenging using AdamW. For usual learning rates, the loss spikes randomly during training, largely lowering the final performance. See Figure 4 for more details. For very small learning rates, the training loss doesn't decrease fast enough to get results in a reasonable amount of time. In this case, we found using SING with AdamW to be a good choice as the normalization prevents the gradient from exploding during training. As a result, the combination of AdamW and SING outperformed AdamW by a large margin. The larger the assessed model, the worse the instabilities. ViT-big [13] does not converge when using AdamW. We tried several sets of hyper-parameters to draw this conclusion.

We used the same hyper-parameter tuning strategy and learning rate scheduler as for ImageNet-1K. The network was trained for 100 epochs using a batch size of 512. The loss we minimized was the MSE. See Table 2 for the results and an ablation study. The ablation study shows that each component of SING helps to achieve a higher learning rate and therefore higher performance. Notably, softplus [40] seems to largely help SING while it is detrimental for AdamW. The normalization seems to be a determining factor for reaching convergence although it does not fully explain the success of SING. We also studied the impact of SING when combined with other optimizers. The results are visible in Table 3. We used all methods with their default hyper-parameters except for SGD where we tried different values for the momentum. We see that for three out of the four assessed optimizers, the variant with SING significantly outperforms its counterpart. For AdaFactor [37] there is barely any performance gain. We claim this is due to the numerous tweaks within the optimizer that have been tailored for a gradient descent without SING.

## 5.3 Natural language processing

In this section, we evaluate the performance of our optimizer on natural language processing tasks. First, we trained a Transformer with pre-norm convention [45] on the IWSLT14 German-to-English (De-En) dataset [4] using the FAIRSEQ [31] library. We used the code of AdaHessian [48] as is but surprisingly we were not able to reproduce the results reported for AdamW. Instead, we used the hyper-parameters reported in [46] and found them to be better, but still below the announced results. Then, we used Hugging Face TRANSFORMERS library [44] to fine-tune Bert [11] on the SQuAD dataset [35] and RoBERTa on SWAG [53]. The results are reported in Table 4. In all cases, the combination of AdamW and SING outperforms a well-tuned AdamW.

|  |  |  | AdamW [26] | AdamW + SING |
|---|---|---|---|---|
| IWSLT14 | From scratch | Transformer [43] | 34.76 | **35.41** |
| SQuAD[1] | Fine-tuning | Bert [11] | 80.53% / **88.39%** | **81.00%** / 88.34% |
| SWAG[1] | Fine-tuning | RoBERTa [25] | 80.45% | **83.33%** |

Table 4: First line: BLUE score on the IWSL14 task, when training a SMALL Transformer [31] from scratch. Second line: Fine-tuning results on the SQuAD dataset [35]; the reported values are the proportion of exact matches and the F1 score. Third line: Fine-tuning results on the SWAG dataset [53]; the reported value is the accuracy.

We noticed that the gradient norm was increasing throughout training. After investigation, it turned out the culprits were the weights and biases of the Layer Normalization [2] layers. We decided to disable the learning of these weights and found the performance of both optimizers to be improved. We claim doing so is not problematic in practice as disabling the learning of these parameters have been pointed out as beneficial in the literature [47].

## 6    Limitations

We tried SING on other tasks such as image denoising, where the models trained with SING attained the same performance of AdamW, but did not result in improved results. This suggests SING's effectiveness can vary depending on the task and architecture. Additionally, we found our optimizer to not work well when used in conjunction with LayerNorm [2] or LayerScale [41]. While a simple fix is to disable the learning of these weights, it raises the question of why and calls for a better solution. Finally, we propose a convergence proof of the iterates defined in 2, which does not incorporate AdamW even though we mainly used their combination in the paper.

## 7    Conclusion

We introduced SING, a plug-and-play technique that improves the stability and generalization of the Adam(W) optimizer in deep learning, and can be used with any optimizer. By leveraging layer-wise gradient standardization, SING enhances the performance of the optimizer without introducing additional hyperparameters. Extensive experimentation across various tasks demonstrates its effectiveness compared to the original AdamW optimizer. Theoretical analysis reveals that SING enables the optimizer to escape narrow local minima within a single step, with the required learning rate inversely proportional to the network's depth. This compatibility with deep neural networks highlights its practicality. The analysis also provides valuable insights into the behavior of SING, such as its convergence rate or stability, and its advantages over traditional optimization techniques.

In conclusion, our proposed SING technique offers a practical and effective upgrade to the Adam(W) optimizer in deep learning. By enhancing stability and generalization, it contributes to the improvement of optimization algorithms in neural network training. The results of our research, combined with the theoretical analysis, open avenues for further exploration, including investigating the compatibility of SING with other optimization frameworks and addressing the challenges associated with specific normalization techniques.

---

[1]We took the code of Hugging Face (https://huggingface.co/transformers/v2.3.0/examples.html) as is and launched it, and found the performance to be lower than announced. More details in Appendix.

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
