# OpenReview forum: "SING: A Plug-and-Play DNN Learning Technique"
_NeurIPS.cc/2023/Conference — Submitted to NeurIPS 2023_

### Official Review · Reviewer_hVDH · 2023-07-04

**Soundness:** 3 good
**Presentation:** 3 good
**Contribution:** 2 fair
**Rating:** 4
**Confidence:** 4

**Summary:**

The paper proposes SING, a simple gradient preprocessing technique that, combined with any optimizer of choice, argues for improved stability and generalization. The paper further provides a theoretical convergence analysis of the approach

**Strengths:**

1. The paper is clear
2. The technique is simple and easy to implement.


**Weaknesses:**

Overall the paper proposes a straight forward extension to the gradient centralization method, where the gradients are also normalized in a pointwise fashion as in other adaptive techniques. The main weaknesses of the paper are:

1. Incremental - i do not think the contribution of this paper merits publication due to its incremental nature. Adaptive optimizers already normalize gradients in a similar way and it is not clear what is added in the proposed method.

2. Non-convincing experiments - Only 1 experiment compares gradient centralization + AdamW (GC + AdamW), which was proposed in [1], which is the closest method to the one proposed in the paper.  By that experiment GC + AdamW already achieves approximately the same performance, hence stripping SING from any practical significance. I do not understand why GC + AdamW is not used as a baseline for other experiments as it clearly shows strong performance. At it stands, it is not clear whether the apparent improvement of SING stems from the GC part of SING, or the added normalization which constitutes it novelty. I would encourage the authors to add this ablation study to the paper to make it more convincing. Finally, results in the paper do not include standard deviation which is be a must for an empirical paper.

3. The theory can be equally applied to other adaptive optimizers, hence it is not special to SING




[1] - Yong et al - "Gradient Centralization: A New Optimization
Technique for Deep Neural Networks"

**Questions:**

I would appreciate if the authors could clarify what is the motivation to normalize the gradients element-wise when this is already done in any adaptive technique (in various forms that might not match SING exactly)

**Limitations:**

Limitations are adequately addressed.

---

> ### Author Rebuttal · Authors · 2023-08-08
>
> First off, we would like to thank the reviewer for their time and feedback. We hope our answers will help clear up some misunderstandings.
>
> 1) **About the incremental aspect and the difference with adaptive techniques**
>
> The novelty of this paper comes from a novel combination of existing ideas, the theoretical analysis, and the demonstration of the positive impact of the combination on the results. Notably, in the ablation study of Table 2, we show that each individual previous method (GC, normalization, LookAhead, Softplus) is outperformed by a significant margin by the proposed combination.
>
> We respectfully disagree with the reviewer about the fact that adaptive optimizers already normalize the gradient in a similar way.
> In fact, adaptive optimizers such as Adam(W) normalize the gradient element-wise, and the normalization factor is computed via a temporal average. On the other hand, SING normalizes the gradient layer-wise with an instantaneous normalization factor. This is fairly important because, as pointed out in point 3) of the general comment, the temporal mean computed in Adam(W) cannot filter out a sudden increase in the gradient magnitude. This incapability is the cause of the instability observed in Figure 4 of the paper.
> On the other hand, the normalization operated in SING ensures the gradient magnitude remains constant layer-wise, preventing any such kind of explosion.
>
> Furthermore, the normalization of SING is compatible and can work hand-in-hand with adaptive methods such as Adam. Indeed, SING normalizes the gradient layer-wise, and Adam does so element-wise. Therefore, if one weight of a given layer has a small magnitude compared to the others, SING won’t be able to increase its value, but Adam will. This explains the success of the mixture of AdamW + SING, as pointed out in Tables 1, 2, 3, and 4.
>
> Reviewer JvsR also raised questions about the interplay between SING and AdamW. Please see also our answer (3) to reviewer JvsR, where we proposed to add a discussion about this to the paper.
>
> 2) **Non-convincing experiments**
>
> We believe that the reviewer might have misunderstood the results presented in Tables 1 and 2 and would like to respectfully correct the following claims made in his review:
>
> *"Only 1 experiment compares gradient centralization + AdamW (GC + AdamW)."* \
> Two experiments compare AdamW+SING with AdamW+GC: Table 1 (ImageNet) and Table 2 (Ablation study on the Rectangle Depth Estimation dataset)
>
> *"By that experiment, GC + AdamW already achieves approximately the same performance, hence stripping SING from any practical significance. I do not understand why GC + AdamW is not used as a baseline for other experiments as it clearly shows strong performance."* \
> Based on the results of Tables 1 and 2, AdamW+CG has systematically worse performance than AdamW, and even more when compared against AdamW+SING.
>
> The results in the ablation study of Table 2 suggest that normalization is the key factor making it work and not GC. In more detail, simply adding the normalization to AdamW improves the accuracy by +14.87%, whereas adding GC to AdamW reduces it by -0.67%. From this +14.87% of improvement of SING, GC only contributes to 2.70% (*i.e.* SING with only normalization and without GC still improves AdamW by 12.17%). Based on our experience, we found that GC is mainly useful to allow for a larger learning rate, therefore escaping more local minima, as per Theorem 3.1.
>
> These results lead to the conclusion that SING cannot be interpreted as a *“straightforward extension of the gradient centralization paper”*.
>
> 3) **About standard deviations**
>
> We agree with the reviewer. We did not compute standard deviations on ImageNet mainly due to our limited access to computational resources. However, please note that we did include standard deviations on CIFAR 100 in the supplementary material. We also want to stress that the results reported in the paper are not cherry-picked. Except for CIFAR100, once we defined our methodology for hyper-parameter tuning, we ran each training once and reported the results obtained. This reduces the chances of bias in the results.
>
> In addition, we added in Table 1 of the PDF accompanying this rebuttal the standard deviations for the first line of Table 2, where it can be seen that the trainings with AdamW always explode whereas SING is always stable, which is the main claim of the paper. We will add these standard deviations (and the ones for the rest of Table 2) in the final version of the paper.
>
> 4) **The theory could be applied to other adaptive optimizers**
>
> We respectfully disagree with the reviewer on this point as well. We insist that the layer-wise normalization of SING and the element-wise temporal normalization of adaptive optimizers such as Adam are not equivalent. To the best of our understanding, none of the theory developed in the paper could be applied to other adaptive optimizers. Take Theorem 3.1, for example. It relies on the fact that SING takes steps with constant size due to normalization. Adaptive optimizers do not have this property. For adaptive optimizers, the size of the steps can be altered easily, as described in the third part of the global comment.
>
> **Q1: I would appreciate it if the authors could clarify what is the motivation to normalize the gradients element-wise when this is already done in any adaptive technique (in various forms that might not match SING exactly)**
>
> The normalization applied in SING is not element-wise but layer-wise, and most importantly, it is instantaneous and not a temporal running average. The motivation is to normalize the gradient so as to catch any pathological case (such as exploding and vanishing gradients) so that it does not interfere with the temporal statistics of adaptive methods. This normalization also avoids explosions such as the one depicted in Figure 4 of the paper and point 3) of the general comment.

---

> > ### Comment · Reviewer_hVDH · 2023-08-21
> >
> > I thank the authors for their clarifications, which have alleviated some of my concerns.
> > However, I will keep my score due to incremental novelty and inadequate experimental validation for a practical paper. The later is especially concerning, given that the authors propose a general purpose optimizer, without much evidence for its performance in medium to large scale experiments. My suggestion for the authors is to gather more experimental evidence for their claims (not on CIFAR) and resubmit to another venue.

---

> > > ### Author Response · Authors · 2023-08-21
> > >
> > > We thank the reviewer for his reply. The reviewer mentions that our experimental validation is inadequate (suggesting it is limited to CIFAR).
> > >
> > > We would like to clarify that in addition to CIFAR100 the paper also features evidence about the better performance of SING on *"medium to large scale datasets"* on several tasks, as it features experiments on
> > > - Image classification on ImageNet (Section 5.1, Table 1)
> > > - Text translation on IWSLT14 (Section 5.3, table 4)
> > > - Question answering on SQuAD (Section 5.3, Table 4)
> > > - Multiple choices question answering on SWAG (Section 5.3, Table 4)
> > >
> > > We also use a wide range of architectures ranging from residual networks to transformers.
> > >
> > > In addition, we would like to point out that this is a new remark and was not included as a weakness in the initial review, nor by the other reviewers, which on the contrary, had positive comments regarding the variety/scale of the numerical experiments [Ahr6, QCf2, JvsR, CMfs] even including it among the paper strengths [Ahr6, QCf2, JvsR].

---

### Official Review · Reviewer_CMfs · 2023-07-04

**Soundness:** 2 fair
**Presentation:** 2 fair
**Contribution:** 2 fair
**Rating:** 6
**Confidence:** 3

**Summary:**

The paper proposed a simple and hyper-parameters-free way to improve the stabilization and generalization properties of optimizers used in deep-learning scenarios. They show that with gradient centralization and gradient normalization methods, SING can escape the local minima with large step sizes theoretically. The authors provide several experiments on datasets like ImageNet-1K, RDE, and some NLP tasks to show the superiority of SING together with popular optimizers like AdamW. They also give out some other theoretical results like convergence and invariance properties.

**Strengths:**

1. The experiments on real datasets show that SING+AdamW performs significantly better than other baselines at image classification, depth estimation, and NLP tasks. This efficient method is also simple and not requires additional hyper-parameters.
2. The authors show that SING can escape the basin of attraction of the critical point when the step size is sufficiently large, and the stepsize threshold is inversely proportional to the network's depth, while GD cannot. And the experiments result (Figure 4) show that SING can stabilize the performance of the optimizers like AdamW.
3. The paper is well-writen and easy to follow.

**Weaknesses:**

1. Although the empirical results are remarkable, the novelty of this paper is limited. As mentioned in the paper, gradient centralization[1] and gradient normalization[2,3,4] are common methods in the previous works, and this paper combines these two methods and systematically investigates the properties of SING.
2. The theoretical analysis just focuses on the gradient rather than combining it with momentum and scheduler, but since the gradient is normalized and the step size is large, the momentum and learning rate scheduler is critical for the global convergence, like in Thm 3.3, 3.4, $\eta$ is small, but $\eta$ needs to be large to escape local minima from Thm 3.1.
3. Thm 3.3, 3.4 requires that the mini-batch size B be some concrete value, this is too strict and it's better to relax this assumption.

[1] Hongwei Yong, Jianqiang Huang, Xiansheng Hua, and Lei Zhang. Gradient centralization: A new optimization technique for deep neural networks. In Computer Vision–ECCV 2020: 16th European Conference, Glasgow, UK, August 23–28, 2020, Proceedings, Part I 16, pages 635–652. Springer, 2020
[2] Ashok Cutkosky and Harsh Mehta. Momentum improves normalized sgd. In International conference on machine learning, pages 2260–2268. PMLR, 2020.
[3] Ryan Murray, Brian Swenson, and Soummya Kar. Revisiting normalized gradient descent: Fast evasion of saddle points. IEEE Transactions on Automatic Control, 64(11):4818–4824, 2019.
[4] Shen-Yi Zhao, Yin-Peng Xie, and Wu-Jun Li. On the convergence and improvement of stochastic normalized gradient descent. Science China Information Sciences, 64:1–13, 2021.

**Questions:**

1. The results in figure 3 seem not fully converged, can the author propose the results with larger total training epochs?
2. If the author can relax the assumption of the mini-batch size B in Thm 3.3 and 3.4, similar to weakness-3?
3. The W(x) in definition 3.1 should be A(x) in the later paper?

**Limitations:**

The author mentions that their method cannot be used together with LayerNorm or LayerScale, and the theoretical results not incorperate with AdamW.
I don't find ethical or immediate negative societal consequences in this work.

---

> ### Author Rebuttal · Authors · 2023-08-08
>
> First, we would like to thank the reviewer for reviewing our submission.
>
> 1) **About the limited novelty**
>
> The novelty of our work indeed comes from the combination of multiple ideas but also from the theory developed to explain the behavior of such a combination.
> The relevance of the proposed combination is demonstrated in the ablation study (Table 2 of the paper) since each individual previous method works worse than the proposed method.
>
> Furthermore, we utilize the combination to tackle the problem of training stability, which is also novel because none of these existing methods tackle it (*i.e.* all were introduced for other goals).
>
> 2) **The theoretical analysis just focuses on the gradient rather than combining it with momentum and scheduler, but since the gradient is normalized and the step size is large, the momentum and learning rate scheduler is critical for the global convergence, like in learning rate in Thm 3.3, 3.4, is small, but it needs to be large to escape local minima from Thm 3.1.**
>
> Most optimizers used in practice suffer from similar trade-offs, which is why learning rate scheduling is used. SING makes it easier to tune the hyper-parameters, as the normalization gives some independence with respect to the energy landscape (Theorem 3.2). SING is to be used exactly like any other optimizer regarding scheduling: a warmup followed by a high learning rate that is decayed over time.
> We refer the reviewer to point 4) of the global comment for more details about our theoretical analysis.
>
> **3) + Q2: Thm 3.3, 3.4 requires that the mini-batch size B be some concrete value, this is too strict and it's better to relax this assumption.**
>
> Unfortunately, a condition on the batch size is necessary with the set of hypotheses we considered. These hypotheses and conditions of the batch size are not unusual (see *e.g.* [1]).
> The intuition is that in the non-convex stochastic case if one wants to reach a more precise solution, one must reduce the variance of the gradient. To do so, the batch size must be increased, which aligns with the common rule of thumbs in the deep learning community that larger batch size is preferable to achieve better training. The same reasoning works for the learning rate.
>
> However, we do not think these hypotheses precisely capture the entire structure of neural networks; see [4] for more details. We refer the review to point 4) of the general comment for a longer discussion on the theoretical results.
>
> **Q1: The results in figure 3 seem not fully converged, can the author propose the results with larger total training epochs?**
>
> The results of Figure 3 are typical for a cosine decay scheduling. Increasing the number of epochs would widen the plot and wouldn’t change its aspect.
> Cosine decay is widely used in the literature nowadays [2,3], and we carefully verified it improved the performance for all the assessed optimizers before using it. See point 6) of the answer to reviewer JvsR for more details.
>
> **Q3: The W(x) in definition 3.1 should be A(x) in the later paper?**
>
> Thank you for pointing it out. As stated in the global comment, we will make sure to fix it.
>
> [1] Zhao, S. Y., Xie, Y. P., & Li, W. J. (2021). On the convergence and improvement of stochastic normalized gradient descent. Science China Information Sciences, 64, 1-13. \
> [2] Liu, Z., Mao, H., Wu, C. Y., Feichtenhofer, C., Darrell, T., & Xie, S. (2022). A convnet for the 2020s. In Proceedings of the IEEE/CVF conference on computer vision and pattern recognition (pp. 11976-11986). \
> [3] Liu, Z., Lin, Y., Cao, Y., Hu, H., Wei, Y., Zhang, Z., ... & Guo, B. (2021). Swin transformer: Hierarchical vision transformer using shifted windows. In Proceedings of the IEEE/CVF international conference on computer vision (pp. 10012-10022). \
> [4] Ma, S., Bassily, R., & Belkin, M. (2018, July). The power of interpolation: Understanding the effectiveness of SGD in modern over-parametrized learning. In International Conference on Machine Learning (pp. 3325-3334). PMLR.

---

> > ### Comment · Reviewer_CMfs · 2023-08-18
> >
> > Thank you for your reply, I think it addresses most of my concerns. I will raise my score and wait for the response from other reviewers.

---

### Official Review · Reviewer_Ahr6 · 2023-07-05

**Soundness:** 3 good
**Presentation:** 3 good
**Contribution:** 2 fair
**Rating:** 7
**Confidence:** 4

**Summary:**

In this paper, authors have proposed a method (called SING) for stabilizing the optimization algorithms used in training of deep models. The proposed method is based on only a layer-wise standardization of the gradients without introducing any additional hyper-parameters. In addition, a theoretical analysis for convergence to a stationary point is provided. Extensive empirical simulations show improvement of the training performance when the existing optimization algorithms use the proposed approach on various tasks.

**Strengths:**

In general, the paper is well-written, and concepts have been presented in an accessible way. Providing theoretical results, including the convergence and invariance properties provide more credibility to the proposed method. Furthermore, experiments on different architectures and on various datasets is another strength point of this paper.

**Weaknesses:**

I need some clarifications on the followings:

1 - As mentioned in the theorems 3.3 and 3.4, convergence is guaranteed only to a stationary point. On the other hand, Theorem 3.1 states that the algorithm can escape from a narrow local minimum. How can SING guarantee that the stationary point is a local minimum (what happens if the algorithm converges to a saddle point or even local maximum) ?

2 - What defines the narrow local minimum and the wide local minimum. There is no curvature information/notion in Theorem 3.1 to distinguish local flat minimum from the sharp one.

3 - In Theorem 3.3, $\epsilon^2$ is given by $\sigma^2/B$, so to have an arbitrary small error on the expectation of the gradient at some stationary point, $\sigma$ should scale as $\mathcal{O}(\frac{\sqrt{B}}{D})$. My question is for a very large model, (i.e., $D$ is huge), does the assumption (8) hold for every $x\in\mathbb{R}^p$? I am not sure how the assumption holds for a highly non-convex loss in a large deep model? This assumption is stronger assumption than other approaches. Typically, ADAM, and other optimization in deep learning either assume some level of convexity or use somehow reasonable assumptions like small gradient, or bounded sequence of estimates, etc.,

4 - Regrading the previous point, it is a good idea to run an experiment to illustrate the effect of $D$ on the training performance with SING.

5 - Do the results in experiment section (Table 1, 2, and 3) show the validation accuracy or the training accuracy ? Please clarify this.

6 -The convergence result doesn't provide any insight for the generalization to the unseen data. It is a purely an optimization perspective.

**Questions:**

The largest ball contained within the basin of attraction in Definition 1.1 is denoted by $\mathcal{B}$; however, in other places, authors use $A()$, am I right?

I couldn't find the definition of $(\epsilon, \phi)$-stationary point used in Theorem 3.4.

**Limitations:**

Please see my comments for Weaknesses and Questions.

---

> ### Author Rebuttal · Authors · 2023-08-08
>
> First off, we would like to thank you for your time and comments, and we hope that our answers will help clarify the paper.
>
> 1) **Convergence to a stationary point**
>
> As pointed out in Definition 3.1, for a saddle point $\mathcal{B}(x^*) = \lbrace x^*\rbrace$ hence $r=0$, and therefore Theorem 3.1 guarantees that the saddle point is escaped in exactly one iteration.
> Take for instance $f(x) = x^3$, in this case $x^* = 0$, $W(x^*) = \mathbb{R}\_{-}$ and the largest ball centered around $x^*$ contained within $W(x^)$ is $\lbrace x^* \rbrace$. In practice, gradient descent will never converge to that point except if you fall exactly on it, which has a probability $0$ of happening.
> The same goes for local maxima: in this case $W(x^*) = \lbrace x^* \rbrace$ and $B(x^*) = \lbrace x^*\rbrace$. For example, take $f(x) = -x^2$, in this case $x^* = 0$, $W(x^*) = \lbrace 0\rbrace$ and $\mathcal{B}(x^*) = \lbrace 0\rbrace$.
> For more details, we refer the reviewer to [1].
>
> 2) **Narrow vs wide and flat vs sharp**
>
> A narrow local minimum corresponds to a point $x^*$ such that $\mathcal{B}(x^*)$ has a small radius. Conversely, if the radius is high, it is considered as being wide. We agree with the reviewer that this terminology is unclear and will add clarification to the paper.
> The local sharpness, however, is contained within the gradient norm. As can be seen in Equation (5), the flatter the local minimum (and hence the lower the gradient norm), the harder it is for SGD to escape. As SING is independent of the gradient norm, it is independent of the sharpness and hence escapes any local minimum, provided it is narrow enough.
> Lastly, sharpness/flatness is an easy-to-understand notion in 2 or 3D but is harder to manipulate and understand in higher dimensions. Indeed, the sharpness is a local information, and a local minimum could be flat and sharp. That is why we decided not to introduce and manipulate this notion in more detail.
>
> 3) **Assumption (8)**
>
> The variable $\sigma$ measures how well the stochastic gradient approximates the real gradient. It is a constant that is more likely to depend on the dataset than on the network’s architecture as long as the gradient is regular enough (Assumption (9)). Assumption (8) is a very classical assumption made in non-convex optimization in the stochastic setting (see *e.g.* [2, 9]).
> We respectfully disagree with the reviewer that this “assumption is stronger than other approaches” when other approaches consider a deterministic convex setting, and we consider a non-convex stochastic one.
> If necessary, assumption (8) could be replaced by $\forall t, \mathbb{E}[\|\nabla F(x\_t) - \nabla f(x\_t)\|\_2^2] \leq \sigma^2$ as it is only used in this case.
>
> 4) **Effect of D**
>
> While we agree it would have been interesting, such an experiment would have been hard to conduct in a principled and rigorous way as typical models only exist in three to five sizes.
> A (limited) version of such a study can be seen in Table 1, where ResNet18 and ResNet34 are evaluated on ImageNet when ResNet18 has $D=62$ and ResNet34 has $D=110$.
>
> 5) **Training vs validation accuracy**
>
> In every table of this paper, the validation accuracy is reported. The only time a training metric is reported is in Figure 4 to better highlight the explosion in the training loss.
> We thank the reviewer for pointing this ambiguity out and will clarify this in the paper.
>
> 6) **The convergence result doesn't provide any insight for the generalization to the unseen data. It is a purely an optimization perspective.**
>
> The question of generalization is an open problem, and the community does not always agree on a correct definition [8]. It is also largely a statistical problem that is outside the scope of this paper. However, given the apparent link between the width of the local minima and generalization [3,4,5,6,7], Theorem 3.1 suggests that by controlling the learning rate, SING is more likely to skip narrow local minima and generalize better.
>
> **Q1 - The largest ball contained within the basin of attraction in Definition 1.1 is denoted by B; however, in other places, authors use A, am I right?**
>
> Thank you for pointing it out. As pointed out in the global comment, we will make sure to fix it.
>
> **Q2  - I couldn't find the definition of $(\epsilon, \phi)$-stationary point used in Theorem 3.4.**
>
> It is defined on lines 163-164: it is a point such that in the limit $\epsilon \to 0$, the norm of the gradient converges to a point in $\text{Ker}(\phi)$.
>
> [1] Murray, R., Swenson, B., & Kar, S. (2019). Revisiting normalized gradient descent: Fast evasion of saddle points. IEEE Transactions on Automatic Control, 64(11), 4818-4824. \
> [2] Zhao, S. Y., Xie, Y. P., & Li, W. J. (2021). On the convergence and improvement of stochastic normalized gradient descent. Science China Information Sciences, 64, 1-13. \
> [3] Cooper, Y. (2018). The loss landscape of overparameterized neural networks. arXiv preprint arXiv:1804.10200. \
> [4] Goodfellow, I. J., Vinyals, O., & Saxe, A. M. (2014). Qualitatively characterizing neural network optimization problems. arXiv preprint arXiv:1412.6544. \
> [5] He, H., Huang, G., & Yuan, Y. (2019). Asymmetric valleys: Beyond sharp and flat local minima. Advances in neural information processing systems, 32. \
> [6] Keskar, N. S., Mudigere, D., Nocedal, J., Smelyanskiy, M., & Tang, P. T. P. (2016). On large-batch training for deep learning: Generalization gap and sharp minima. arXiv preprint arXiv:1609.04836. \
> [7] Pennington, J., & Bahri, Y. (2017, July). Geometry of neural network loss surfaces via random matrix theory. In International conference on machine learning (pp. 2798-2806). PMLR. \
> [8] Zhang, C., Bengio, S., Hardt, M., Recht, B., & Vinyals, O. (2021). Understanding deep learning (still) requires rethinking generalization. Communications of the ACM, 64(3), 107-115. \
> [9] Bottou, L., Curtis, F. E., & Nocedal, J. (2018). Optimization methods for large-scale machine learning. SIAM review, 60(2), 223-311

---

### Official Review · Reviewer_QCf2 · 2023-07-07

**Soundness:** 3 good
**Presentation:** 3 good
**Contribution:** 3 good
**Rating:** 6
**Confidence:** 4

**Summary:**

The paper presents SING (StabIlized and Normalized Gradient), a new method designed to enhance the stability and generalization capabilities of the Adam(W) optimizer. SING involves a layer-wise standardization of the gradients that are input into Adam(W), and does not require the introduction of additional hyper-parameters. This makes it straightforward to implement and computationally efficient.

The authors demonstrate the effectiveness and practicality of SING through improved results across a broad range of architectures and problems, including image classification, depth estimation, and natural language processing. It also works well in combination with other optimizers.

In addition to these experimental results, a theoretical analysis of the convergence of the SING method is provided. The authors argue that due to the standardization process, SING has the ability to escape local minima narrower than a certain threshold, which is inversely proportional to the depth of the network. This suggests that SING may offer significant advantages in training deep neural networks.

**Strengths:**

1. As the authors have claimed, the proposed method can be applied in a plug-and-play style with impressive applicability to a lot of tasks, datasets and optimizers. Without additional hyperparameters introduced, I think this work has great potential to become a standardized training technique with big impact in the community. And the authors did provide the elegantly implemented source code in PyTorch which I think is already close to ready to be included in the standard PyTorch library.

2. Empirical performance is impressive with huge improvements over baseline optimizers in many settings.

3. Mostly the paper is written in quality and easy to follow with only a few ambiguities, which I will mention in the weaknesses part.

**Weaknesses:**

1. Firstly, I believe there is a misalignment between the theoretical analysis and practical method. To be specific, the analysis in Theorem 3.1 compares the learning rate needed for escaping local minima for SING and SGD. However, as the authors claimed previously, the SING algorithm is proposed to overcome the limitations of Adam(W). Therefore, it would be better to directly analyze SING against Adam(W), which is also mainly compared against in the experiment part.

2. Some issues in terms of writing. One is that the authors did not formally formulate the centralize operation in math equations but only in codes, which could be confusing for readers who are not familiar with PyTorch framework. I strongly recommend the authors to provide strict math formulations instead of ambiguous codes only. For example, at least I am still confused the mean operation is executed over which dimension and what the meaning is for that averaging. A second issue in writing is that it seems the authors interchangeably use the terms "learning rate" and "step size" in Section 1 but treat them as different things in Section 3. I hope the authors could clarify the differences or use one term consistently.

**Questions:**

1. According Figure 4, the spikes occur once during one training process. Do the authors have any comments on what causes the spikes exactly?

2. The second row of Table 2 seems to be wrongly presented.

3. What if the momentum also comes into the picture to be combined with SING?

---

> ### Author Rebuttal · Authors · 2023-08-08
>
> First off, we would like to thank you for your kind comments and your detailed feedback.
>
> 1) **Misalignment between theoretical analysis and practical method**
>
> We refer the reviewer to point 4) of the global comment for more details about our theoretical analysis.
>
> 2) **Confusing definition of the centralization operation**
>
> This is a good point. We did not detail it more in the paper as the operation has already been introduced and thoroughly discussed in [1]. We agree that the definition is ambiguous even in the original paper, and we will try to clarify it in the appendix.
>
> 3) **“learning rate” and “step size”**
>
> We agree that the way these two terms are used is confusing in the paper. We identify as “learning rate” the value $\eta$ defined in Equation (2). We refer to “step size” as being the size of the steps taken by the algorithm *i.e.* if the updates are $x\_{t+1} = x\_{t} - \eta g\_t$ it is defined as $\eta \|g\_t\|\_2$.
> We will make sure to correct the paper to make the difference clearer.
>
> **Q1: According Figure 4, the spikes occur once during one training process. Do the authors have any comments on what causes the spikes exactly?**
>
> We have identified the spike to come from an unexpected rise in the gradient magnitude from one iteration to the other. This recent preprint reports a similar phenomenon on large language models [8]. As pointed out in point 3) of the general comment, the temporal averaging of Adam(W) fails to filter out the sudden increase and creates the spike. The sudden rise is followed by smaller updates which prevent Adam from recovering to the original level. However, the cause of the sudden growth is unknown (an explanation is conjectured in [8]).
> In this case, SING works particularly well because the normalization operation in Equation (2) prevents the gradient magnitude from changing during training.
>
> **Q2: The second row of Table 2 seems to be wrongly presented.**
>
> Table 2 is an ablation study. In more detail, we study what happens if we remove one component from SING and what happens if we add this component to AdamW. In the second row, we see that adding the normalization defined in Equation (2) to AdamW improves its performance. We also see that removing the normalization component from SING prevents it from converging altogether for all the assessed learning rates and weight decays. NC stands for no convergence. The maximum learning rate reported corresponds to the maximum learning rate for SING and hence isn’t reported in this row of Table 2 since it didn’t converge.
> We will modify the description of Table 2 to include a better explanation.
>
> **Q3: What if the momentum also comes into the picture to be combined with SING?**
>
> First, momentum is combined with SING, as shown in Algorithm 2. In fact, any momentum strategy could be used with SING. SING modifies (and standardizes) the gradient passed as input to the optimization algorithm (which can contain momentum).
>
> Secondly, one could think about ways of incorporating momentum within the computation of the gradient norm (or the gradient mean). We argue it would be counterproductive, as temporal averaging would fail to filter out local and sudden increases in the gradient norm. This is what happens in Adam(W) and what we describe in point 3) of the general comment. Hence, doing so is very likely to remove the stability properties of SING.

---

### Official Review · Reviewer_JvsR · 2023-07-17

**Soundness:** 2 fair
**Presentation:** 3 good
**Contribution:** 1 poor
**Rating:** 3
**Confidence:** 4

**Summary:**

The paper proposes SING, a plug-and-play approach to enhance optimizers without introducing additional hyperparameters.

The idea consists of standardizing gradients in a layer-wise manner prior to the host optimizer’s execution, and is motivated by factors such as easier escaping of narrow minima and invariance properties.

Experiments on image classification, depth estimation, and NLP tasks such as NMT and QA are used to assess the proposed method’s performance and measure improvements over host optimizers.


**Strengths:**

The paper is written clearly and well-organized.

The proposed method is easy to implement and works in a plug-and-play fashion. The layer-wise gradient standardization can be viewed as a gradient pre-processing step and hence completely agnostic to the host optimizer, making the approach general and flexible.

The authors provide theoretical analyses to motivate and better understand SING. The convergence analysis considers the smooth non-convex bounded variance setting, which I believe to be a good balance between assumptions and how well it captures network training.

Experiments include different tasks from two domains and distinct network architectures, including ResNets and Transformer-based models.


**Weaknesses:**

The theoretical analysis is not helpful in motivating or better understanding SING’s benefits.

While SING adopts layer-wise normalization, it seems that the analysis holds given any partition of the $p$ many parameters in $D$ many sets – i.e. the fact that each of the $D$ tensors is assumed to correspond to a different layer is not necessary nor used anywhere in the analysis. We can then consider the effect of variable $D$ for a fixed $p$ (grouping the parameters in larger or smaller sets, say filter-wise, kernel-wise, or even parameter-wise), and we recover Normalized Gradient Descent (NGD) with $D=1$ and a form of sign SGD with $D=p$ (this has been studied in previous works to understand how normalizing gradients of coarse/fine-grained parameter sets can affect performance and convergence).

This has two concerning implications:

- For Theorem 3.1

Since $||\frac{g}{\Gamma (g)}|| = \sqrt{D}$, it follows that the post-processed gradients will scale (in norm) as $\sqrt{D}$, and hence $\eta_{SING}$ in Theorem 3.1 is fundamentally ‘undoing’ this scaling, hence claiming that ‘SING can escape local minima narrower than a threshold that is inversely proportional to the network’s depth’ is not very meaningful.

A similar argument would be to pre-process the gradient by scaling it up by 100 and claiming that now we can use a 100x smaller learning rate, which, although technically correct, is not useful.

While I’m aware that the layer-wise normalization will actually change the update direction and not just scale it, it seems that this change in direction does not play any positive role in the presented analysis.

Finally, one could simply set $D=p$ (i.e. artificially view each parameter as an independent layer) and Theorem 3.1 would state that the sufficient learning rate to escape narrow minima would actually decrease way more aggressively (as 1/sqrt(# parameters) instead of 1/sqrt(# layers)) – this is clearly not actually useful since we’re just scaling up the (norm of) post-processed gradients (compared to the layer-wise case) and compensating by scaling down the learning rate.


- For Theorems 3.3. and 3.4

To achieve stationarity $\delta$ independent of $D$ (i.e. $\epsilon \propto \frac{\delta}{D}$) we would set $\eta = \Theta(\frac{\delta}{D})$, $T = \Theta(\frac{D^2}{\delta^2})$, $B = \Theta(\frac{D^2}{\delta^2})$, where we are ignoring dependencies on $F(x_0)$ and $L$.

This means that, in the original case where $D =$ # layers, the guarantee requires both the number of iterations and the batch size to increase quadratically with the depth of the model, which is concerning.

Moreover, if we set $D=1$ (i.e. NGD) we actually minimize the required number of iterations and batch size. Therefore, these results do not motivate or support layer-wise normalization, and actually question this design choice by offering significantly better guarantees for NGD.

- Other points

Although the theoretical analysis considers updates following Eq. 2, the experimental studies heavily focus on AdamW + SING, which is not well-discussed. My main concern in this case is that the normalization from SING affects both $m_t$ and $v_t$ in the numerator and denominator of AdamW, respectively. It is unclear what is really happening in this case.


It seems that for long enough training time windows, if the layer-wise gradient norms remain roughly constant then the normalization effect would cancel out, reducing to AdamW’s updates (that is, if the $\epsilon$ term in the denominator of AdamW is negligible). Accounting for $\epsilon$, on the other hand, yields AdamW’s updates except with different values for $\epsilon$ for each layer, each scaled by the layer’s gradient norm.

Although it is unlikely that layer-wise gradient norms remain roughly constant for many enough iterations, this hints that SING’s normalization might be affecting the size of $\sqrt(v_t)$ compared to $\epsilon$ differently for each layer. This could result in confounding effects since the value of $\epsilon$ (compared to $\sqrt(v_t)$) can play a major role in the behavior of Adam-like methods, potentially improving the performance in multiple settings.

The experimental analysis could also be substantially improved. The hyperparameter tuning strategy (choosing best learning rate, then fixing it to choose best weight decay) can easily lead to suboptimal values, especially for SGD and any adaptive method that does not inherently incorporate AdamW’s weight decay decoupling (see Fig. 1 and 2 of Loshchilov & Hutter).

This can lead to an unfair advantage to AdamW and AdamW + SING over all other methods. The cosine schedule is also known to improve AdamW’s performance and more often than not harm SGD (compared to step-wise), hence it would be valuable to also collect results with a step-wise schedule for a more comprehensive and clear comparison.

There is also some loss in novelty from the fact that the actual method that plays the key role in the experiments also adopts LookAhead and softplus calibration. In particular, centering is not novel although it has been explored more extensively for the 2nd moment estimate (centered RMSProp, AdaBelief, SDProp, ACProp, etc), and layer-wise gradient normalization for adaptive methods has also been studied (AdaShift & AvaGrad – none of the two are cited or discussed). These methods should be included in the comparison to have a clear picture that would allow a proper assessment of SING.

Finally, there are also concerns regarding the other vision tasks. It is unclear what was the exact ResNet-18 model used for CIFAR-100: if it is a ~11M param model, then it is a wider version (DeVries ResNet-18) which differs from the one originally proposed by He et al. and achieves over 77% acc. on CIFAR-100 when trained with SGD (see LookAhead’s paper and DeVries&Taylor).

This would indicate issues with the CIFAR-100 results in Table 1, since results with SGD would be ~1.4% worse even with additional augmentation and 100 extra training epochs. As for depth estimation, the dataset is synthetic and not well studied, hindering a proper assessment of its results. Nonetheless, ViT’s are typically well-trainable with SGD if warmup and gradient clipping are employed (which is common practice for these models).

The fact that SGD achieved 0.25% accuracy suggests that the experimental setup should be revised – warmup and grad clipping should be adopted for SGD since they are common practice, especially if SGD only achieves 0.25% accuracy without them.


**Questions:**

See my points above regarding points for improvement.

**Limitations:**

The authors discuss limitations satisfactorily.

---

> ### Author Rebuttal · Authors · 2023-08-08
>
> We would like to thank reviewer JvsR for their detailed and insightful review. The reviewer’s feedback allowed us to improve the presentation of the paper. We will make sure to thank them in the final version of the paper. We hope our modifications and answers will help the reviewer reconsider their assessment of the paper.
>
> 1) **No theoretical justification for the layer-wise normalization.**
>
> We agree with the reviewer. We currently have no theoretical analysis showing the advantage of layer-wise normalization. In the paper, we decided to adopt a general setting that is network architecture agnostic. Therefore, the provided arguments apply to any partition of the parameters. The chosen normalization is motivated by intuition and practicality. Take, for instance, the case of a network with a linear layer. SING for a linear layer has two effects: (i) standardizing the inputs to the layer and (ii) removing the magnitude of the partial derivative of the loss.
> The layer-wise normalization is adapted to how current optimizers are implemented, which work layer by layer. Computing the norm across different tensors would require making additional passes through all the layers of the network at every optimization step.
>
> We ran additional experiments to verify if layer-wise partitioning is a good choice. The results are available in Table 2 of the companion PDF.
> The experiments suggest that layer-wise partitioning works better than the alternatives. We will add a paragraph discussing this issue in the paper and add these results in the supplementary material.
>
> 2) **About Theorems 3.3 and 3.4**
>
> Please see point 1) of the general comment. The dependency is now linear instead of quadratic.
>
> 3) **About the mix of AdamW and SING**
>
> Although the complexity of Adam's formula prevents a detailed theoretical study of the mix, our intuition is that the layer-wise normalization of SING and the element-wise temporal normalizations of Adaw(W) work well together.
> Normalizing the gradient layer-wise could shrink the magnitude of some elements within a parameter tensor and prevent these elements from being updated. The element-wise update of Adam reinstates the magnitude of these elements.
> Furthermore, the normalization of the gradient prevents the explosion described in Figure 4 of the paper and in [8]. See point 3) of the global answer for more details.
>
> Furthermore, as pointed out by the reviewer, if the layer-wise gradient norm remains constant, the normalization is rendered useless. We argue that it is actually a good point: when the training procedure is very stable, using SING + AdamW is equivalent to AdamW. In other cases, SING helps to ensure stability during training.
>
> We propose to add these discussions at the end of section 3, noting that more work needs to be done to fully understand the interplay between these two normalizations.
>
> 5) **About the hyper-parameter tuning strategy**
>
> This paper mainly compares against AdamW as it is the de facto algorithm for most deep learning applications [2,3,4,5]. We agree we have not comprehensively tried all the configurations possible for all methods, but we strived to be as accurate as our computation budget allowed.
> Our main focus was two-fold: 1) ensuring a fair comparison between an optimizer and its version with SING, and 2) avoiding choosing a set of hyper-parameters that would have resulted from an overfit on the validation set.
>
> However, we agree that it might not be optimal for every optimizer. For the experiments on ImageNet and NLP however, the hyper-parameters used were the ones reported in the original sources, and while we tried searching for better ones, we haven't succeeded and ended up using them. These hyper-parameters were likely found using much more computations than us and, therefore, could also play to our disadvantage.
>
> 6) **About the learning rate scheduler**
>
> We checked that the cosine scheduler was not harming the performance of SGD before using it. In every case, we found it was at least as good as the stepwise schedule. Using the cosine scheduling with SGD on ImageNet with a ResNet18 gives a final accuracy of 72.36% against 72.25% when using a stepwise scheduler (using the ffcv recipe at https://github.com/libffcv/ffcv-imagenet which does not exactly matches our setting).
> Furthermore, using stepwise scheduling introduces several additional hyper-parameters, complicating the hyper-parameter tuning.
>
> 7) **Comparison against other adaptive methods**
>
> Following the reviewer's suggestion, we evaluated the impact of SING on AdaShift and AvaGrad. Limited computational resources allowed only RDE dataset comparison. The results are available in Table 4 of the companion PDF.
> Although these methods are adaptive, they seem to require gradient clipping to work. This was not the case for Adam. Note that adding SING to these methods largely improves their results without extra hyperparameters. We will report all these results in the supplementary material and mention them in the main paper.
>
> 8) **Concerns about the ResNet18 used**
>
> We used the ResNet18 of the torchvision library, which, according to the documentation, is the implementation of the original ResNet18 by He et al.
>
> 9) **About the trainability of ViT with SGD**
>
> In our experience, we found the ViT to be notoriously hard to train with SGD, in alignment with other claims in the literature [4,6,7].
> Furthermore, all of our experiments feature warmup, as pointed out in line 219 of the manuscript. We launched additional experiments mixing SGD with gradient clipping, and the results are in Table 3 of the companion PDF.
> While it helps the training, the performance is below SGD + SING. We will add these results to the paper. However, gradient clipping introduces an additional hyper-parameter (which might intuitively differ from one layer to another), whereas SING provides stability without any additional hyper-parameter.

---

### Author Rebuttal · Authors · 2023-08-08

First, we thank all the reviewers for their time, consideration, and hard work.

1) **Reparameterization of Gamma [JvsR]**

Following the advice of reviewer JvsR, we decided to modify Equation (3) such that $\Gamma(x)\_i = \sqrt{D} \|x\_{I\_k}\|\_2$ for $k \in [\\!|1,D|\\!]$ and $i \in I\_k$. This modification corresponds to rescaling the normalized gradient norm to become independent of the network's depth. With this reparameterization, we get the interesting property that $\left\|\frac{x}{\Gamma(x)}\right\|\_2 = 1$. This slightly changes the conclusion that “SING can escape local minima narrower than a threshold that is inversely proportional to the network’s depth” to “SING can escape local minima narrower than a threshold that is independent of the network’s architecture”, yet the message remains broadly the same. We believe that this property better captures our intuitive understanding of the good performance of SING.

This reparameterization of $\Gamma$ simplifies Theorem 3.1 where $\eta\_{\text{SING}} \geq 2r$ (getting rid of the $\sqrt{D}$ term, which was the cause of misunderstandings). Similarly, for Theorems 3.3 and 3.4, the dependencies on $D$ are replaced by dependencies in $\sqrt{D}$, leading to a tighter bound. The paper's conclusions remain unaltered, but their justifications are now more concise and clearer, thus making our presentation more impactful and convincing.

Additionally, such a change does not impact Adam(W) in any way since the updates of Adam(W) are invariant to a constant rescaling of the weights. For SGD, the optimal learning rate found must be multiplied by $\sqrt{D}$ to recover the original behavior.

2) **Typo in Section 3.1 [CMfs, Ahr6]**

We corrected the typos reported by the referees, notably where we used $A(x^*)$ instead of $\mathcal{B}(x^*)$. We thank reviewers CMfs and Ahr6 for pointing it out.

3) **About the cause of the spikes and why Adam(W) is sometimes unstable [JvsR, QCf2, Ahr6, CMfs, hVDH]**

Adaptive optimizers such as Adam normalize the gradient element-wise with normalization factors that are computed as temporal averages. This means that if the gradient norm’s magnitude increases by a large factor from one iteration to the other (which happens frequently with transformers, as was reported in https://arxiv.org/abs/2304.09871), the temporal mean won’t be able to filter out the increase right away.

For instance, applying Adam to the sequence $[1, \dots, 1, 100, 1, \dots, 1]$ yields Figure 1 of the PDF document attached to the rebuttal. As the figure shows, the update's magnitude increases and then decreases to an abnormal level. This results in the spike visible in Figure 4 of the paper and explains why AdamW fails to recover after the spike. With SING, the normalization is done based on the instantaneous norm of the layer gradient, preventing any explosion of this kind.

4) **Theoretical results limited to SGD [QCf2, JvsR, CMfs]**

Reviewers QCf2, JvsR, and CMfs pointed out that the theoretical results were limited to SGD. We agree that the current theory does not cover all the aspects of a practical setting (such as *e.g.* varying learning rates or momentum). A proof for Adam(W) would take considerable effort and time and could be the focus of future works. For example, this 30-page paper [1] illustrates how challenging it is to provide a correct proof of convergence of Adam for a constant learning rate. Adapting it to the varying learning rate setting would be the topic of another publication and is outside of the scope of this very submission. Furthermore, to the best of our knowledge, the theoretical properties of AdamW (such as convergence) have not yet been shown. The provided theory is a first step towards a better understanding of SING’s properties, and we still believe that it provides interesting insights into the method.
As for Theorems 3.3 and 3.4, please be aware that these theorems do not, in fact, prove the advantage of the proposed normalization. However, they enable us to check that the method is consistent *i.e.* that its convergence is maintained under reasonable conditions.

**References**

For this rebuttal, we will cite the following papers: \
[1] Défossez, A., Bottou, L., Bach, F., & Usunier, N. (2020). A simple convergence proof of adam and adagrad. arXiv preprint arXiv:2003.02395. \
[2] Liu, Z., Mao, H., Wu, C. Y., Feichtenhofer, C., Darrell, T., & Xie, S. (2022). A convnet for the 2020s. In Proceedings of the IEEE/CVF conference on computer vision and pattern recognition (pp. 11976-11986). \
[3] Liu, Z., Lin, Y., Cao, Y., Hu, H., Wei, Y., Zhang, Z., ... & Guo, B. (2021). Swin transformer: Hierarchical vision transformer using shifted windows. In Proceedings of the IEEE/CVF international conference on computer vision (pp. 10012-10022). \
[4] Touvron, H., Cord, M., Douze, M., Massa, F., Sablayrolles, A., & Jégou, H. (2021, July). Training data-efficient image transformers & distillation through attention. In International conference on machine learning (pp. 10347-10357). PMLR. \
[5] Wang, W., Dai, J., Chen, Z., Huang, Z., Li, Z., Zhu, X., ... & Qiao, Y. (2023). Internimage: Exploring large-scale vision foundation models with deformable convolutions. In Proceedings of the IEEE/CVF Conference on Computer Vision and Pattern Recognition (pp. 14408-14419). \
[6] Xiao, T., Singh, M., Mintun, E., Darrell, T., Dollár, P., & Girshick, R. (2021). Early convolutions help transformers see better. Advances in neural information processing systems, 34, 30392-30400. \
[7] Chen, X., Hsieh, C. J., & Gong, B. (2021). When vision transformers outperform resnets without pre-training or strong data augmentations. arXiv preprint arXiv:2106.01548. \
[8] Molybog, I., Albert, P., Chen, M., DeVito, Z., Esiobu, D., Goyal, N., ... & Zhang, S. (2023). A theory on adam instability in large-scale machine learning. arXiv preprint arXiv:2304.09871.

---

### Decision · Program_Chairs · 2023-09-21

**Decision:**

Reject

**Comment:**

This work proposed  SING to stabilize previous optimizers for network training. SING mainly introduces a layer-wise standardization of the gradients and does not introduce additional hyper-parameters. The authors also provide a convergence analysis of SING.

Two reviewers gave negative comments and emphasized the problems in their theories, e.g., limitation to SGD and error-bound dependence on the network depth, and also nonconvictive and insufficient experiments. I agree with some parts of these limitations, especially for the experiments. I also worked on optimizers for network training, and found that the experiments in this work did not follow the standard settings in most cases, e.g. standard ResNet setting in He Kaiming, and ViT experiments on the RED dataset instead of the standard ImageNet dataset. Indeed, for ViT model on ImageNet, AdamW shows good stability which shows different observations in this work. Based on reviewers' comments and my own experiences, we tend to reject it. The authors could follow these comments and suggestions to improve their work.